# Pregnancies and Gynecological Follow-Up after Solid Organ Transplantation: Experience of a Decade

**DOI:** 10.3390/jcm11164792

**Published:** 2022-08-16

**Authors:** Alice Bedin, Marie Carbonnel, Renaud Snanoudj, Antoine Roux, Sarah Vanlieferinghen, Claire Marchiori, Alexandre Hertig, Catherine Racowsky, Jean-Marc Ayoubi

**Affiliations:** 1Department of Obstetrics and Gynecology, Hospital Foch, 40, Rue Worth, 92150 Suresnes, France; 2Medical School, University of Versailles, Saint-Quentin-en-Yvelines, 55, Avenue de Paris, 78000 Versailles, France; 3Department of Nephrology, Hospital Bicêtre, 94220 Le Kremlin Bicêtre, France; 4Department of Pneumology, Hospital Foch, 40, Rue Worth, 92150 Suresnes, France; 5Department of Nephrology, Hospital Foch, 40, Rue Worth, 92150 Suresnes, France

**Keywords:** pregnancy, organ transplantation, immunosuppression, kidney, lung, gynecology

## Abstract

In recent years, solid organ transplantations, such as kidney or lung grafts, have been performed worldwide with an improvement of quality of life under immunosuppressive therapy and an increase in life expectancy, allowing young women to consider childbearing. In the current study, we conduct a retrospective study in two French centers for kidney and lung transplantations to evaluate the rate and outcomes of pregnancies, contraception and gynecological monitoring for women under 40 years old who underwent solid organ transplantation. Among 210 women, progestin was the most widely used contraceptive method. Of the 210 women, 24 (11.4%) conceived 33 pregnancies of which 25 (75.8%) were planned with an immunosuppressant therapy switch. Of the 33 pregnancies, 7 miscarried (21.2%) and 21 (63.7%) resulted in a live birth with a high rate of pre-eclampsia (50%). No graft rejections were observed during pregnancies. Among the deliveries, 19 were premature (90.5%, mostly due to induced delivery) and the C-section rate was high (52.4%). No particular pathology was identified among newborns. We conclude that pregnancies following solid organ transplantation are feasible, and while they are at an increased risk of pre-eclampsia and prematurity, they should still be permitted with close surveillance by a multidisciplinary care team.

## 1. Introduction

Solid organ transplantation (SOT) is routinely used to treat organ failure. For example, kidney transplantation (KT) is the most cost-effective treatment for end-stage renal failure [1] and lung transplantation (LT) is performed to treat cystic fibrosis [2]. Indeed, SOT represents about 90,000 surgeries annually worldwide and more than 4400 procedures conducted in France in 2020 [3]. However, transplant recipients must undergo immunosuppressive treatment [4], which causes many complications, such as infections, tumors and cardiovascular disease. These patients are also at risk of organ rejection and long-term functional alteration of the graft because of disease recurrence, chronic toxicity of drugs or chronic rejection. Overall, they must be carefully and regularly followed by a multidisciplinary care team to monitor their health and quality of life, not only during the post-surgery period, but also throughout life. The continuing improvement of treatments has resulted in an upward trend in both the quality of life and life expectancy of SOT patients, as evidenced by the current mean five-year survival rate of more than 55% after lung transplantation [5] and more than 75% after kidney transplantation [6].

Among the numerous issues a transplant recipient faces, childbearing is a major one for these women. For many women, having children is a lifelong goal, yet there are several obstacles to overcome in this context, which are compounded by apprehensiveness regarding the life-long morbidity of one parent. Thus, preconception counseling must be personalized to each patient [7], particularly regarding the wide use of fetotoxic (and/or teratogenic) treatments that must be replaced weeks before conception is attempted.

In addition to preconception counseling, a gynecological work-up is required [8] to screen for cervical dysplasia and gynecologic cancers. If pregnancy is to be avoided, an appropriate contraceptive must be identified. In cases in which pregnancy is desired and permitted, the role of the perinatologists extends beyond the well-known pregnancy-associated risks of serious morbidity to mother and child [9,10,11]. Hypertensive disorders, gestational diabetes [12,13,14], preterm delivery [15] and fetal growth restriction are all observed in SOT recipients at frequencies far beyond that of the general population [16]. Therefore, it is essential that gynecologists work in close collaboration with SOT specialists.

The aim of the present study is to describe the mode of contraception used, the incidence of pregnancies, maternal and fetal outcomes and gynecological cancers in recipients who underwent SOT transplantations in two centers in France spanning a decade. In these centers, only lung and kidney transplantations were performed.

## 2. Materials and Methods

This study was approved by the Foch Hospital ethical committee (IRB00012437).

### 2.1. Study Design

We performed a retrospective analysis of all female patients under 40 years of age who underwent kidney transplantations (Department of Nephrology) between 1 January 2010 and 31 December 2019 or lung transplantations (Department of Thoracic Surgery & Lung Transplantation) between 1 January 2010 and 31 December 2018 at Foch Hospital (Suresnes, France); or kidney transplantations (Department of Nephrology) between 1 January 2010 and 31 December 2019 at Bicêtre Hospital (Le Kremlin Bicêtre, France).

### 2.2. Data Collection

Data were collected from each of the hospital’s databases. The following inclusion criteria were applied to identify study patients:Women aged 15 to 40 years at organ transplantation.At least one year after kidney transplantations and two years after lung transplantations, in line with recommendations to reduce possible complications.No objection recorded for inclusion in medical research.

We excluded female patients who died or underwent a transplantectomy during the first year for kidney recipients and the first two years for lung recipients after the initial surgery, as well as those patients with missing data. The sole patient who underwent a combined lung and kidney transplantation was included in the LT group because she underwent a lung transplantation before the kidney transplantation.

The following data were collected: demographic variables (age, BMI, gravidity and parity at transplantation); medical history (including years of dialysis); data regarding the SOT (date of surgery, type, use of live or deceased donor, reason for SOT, combined or isolated SOT, immunosuppressive drugs used) and occurrence of transplant rejection (number and type: cellular, humoral or both). Rejection was recorded when there was a decrease function of the grafted organ. Cellular rejection was confirmed by a CD8+ T-cell-significant interstitial infiltration, commonly diagnosed on biopsy. Humoral rejection was defined by the association between histological lesion, positive c4d immunolabelling and/or positive donor-specific antibodies in blood circulation.

Other data collected were the use of contraception (classified into estrogen+ progestin; progestin: IUD with levonorgestrel, oral tablets of progestin and depo injection; non-hormonal: copper IUD and preservative and others); occurrence of pregnancy and related data (age at pregnancy, interval between transplantation and pregnancy authorization, interval between transplantation and conception, type of immunosuppressant used one year after transplantation, type of conception: spontaneous or assisted reproductive technology (ART), planned/unplanned pregnancy, pregnancy outcome, hypertensive disorders of pregnancy (including gestational hypertension, pre-eclampsia (PE) eclampsia, chronic hypertension and chronic hypertension with superimposed pre-eclampsia), infections, diabetes, fetal outcome, such as birth weight, prematurity, physical abnormalities, week of delivery, mode of and reason for delivery). Pregnancy outcomes investigated were: birth, early miscarriage (before 12 weeks of gestation (WG)), late loss (after 12 WG), medical termination, ectopic pregnancy, molar pregnancy or medical abortion. Pre-eclampsia was defined as the association between hypertension (systolic blood pressure [SBP] ≥ 140 mmHg and/or diastolic blood pressure [DBP] ≥ 90 mmHg) and proteinuria ≥ 0.3 g/24 h after 20 WG. Severe pre-eclampsia was defined by at least one of the following criteria: severe hypertension (SBP ≥ 160 mmHg and/or DBP ≥ 110 mmHg), uncontrollable hypertension, proteinuria ≥ 3 g/24 h, serum creatinine ≥ 90 µmol/L, oliguria ≤ 500 mL/24 h or anuria, thrombocytopenia < 100,000/mm^3^, liver transaminase levels increased two fold above the upper normal limit, severe persistent epigastric and/or right upper quadrant pain, thoracic pain or dyspnea and acute pulmonary edema or neurologic symptoms (severe cephalgia, visual or auditive impairments, hyperreflexia). HELLP syndrome was defined by hemolysis (LDH > 600 UI/L), liver transaminase levels two times the upper limit of normal and low platelet count (<100,000/mm^3^).

Graft functionality was described using timed, forced expiratory volume (FEV1, expressed as a percentage of the predicted value in a patient) for lung recipients and the glomerular filtration rate (GFR, calculated according to CKD EPI creatinine equation, mL/min/1.73 m^2^) for kidney recipients and for which the normal GFR for a young woman is ≥90 mL/min/1.73 m^2^ with the lower limit of the normal FEV1 being 80% in the general population. Pregnancies were usually contra-indicated for women with a respiratory disorder if the FEV1 was under 40%. We reported these indicators at one and two years after the graft transplant for both SOT types, as well as the occurrence of proteinuria just before and during pregnancy and, for pregnant patients, when there was a significant decrease in GFR or FEV1.

### 2.3. Statistical Analyses

Categorical variables were reported as percentages, and means and standard deviations were reported for continuous variables. The unpaired *T*-test and Mann–Whitney test were used to test for significance when variables were continuous, and Fisher’s exact test and the two-sided chi-squared test were used for discrete variables as appropriate. *p* < 0.05 was used to identify differences that were statistically significant.

## 3. Results

As shown in Figure 1, a total of 379 female SOT patients were identified. Of these, 210 met our inclusion criteria, 114 underwent lung transplantation (both lungs) and 96 had a kidney transplantation (one kidney). Reasons for those patients excluded are shown in Figure 1.

### 3.1. Patient Medical Histories and Demographic Characteristics

Patient medical histories and demographic characteristics before transplantation are summarized in Table 1.

The demographic and medical histories of the two transplantations are shown in Table 1. The LT patients were significantly younger than KT recipients at the time of transplantation (27.6 ± 5.6 y versus 31.1 ± 5.9 y, respectively; *p* < 0.001). Of the lung recipients, 48.2% also had a history of diabetes, which was explained by the high majority of patients having cystic fibrosis (91.2%). The remaining LT patients had other respiratory diseases, including primary ciliary dyskinesia (PCD), bronchiectasis and interstitial pulmonary fibrosis (IPF). One had a borderline ovarian tumor treated with ovariectomy before transplantation and most were nulligravid (75.4%) and nulliparous (82.5%).

Kidney recipients were presented with a different profile, compared to LT recipients. Not surprisingly 70.8% had a history of high blood pressure. The major cause of end-stage renal disease was glomerulopathy (26.0%) followed by undetermined nephropathy (19.8%), microvascular condition, diabetes mellitus (13.5%) and hypertension (5.0%), and polycystic kidney disease (5.0%).The average duration of dialysis before transplantation was 2.34 ± 2.2 years. A small majority (51.0%) of these women had been pregnant at least once before transplantation and 44.8% had children. Twelve of the 20 patients (60%) with a known history and who had previously given birth had experienced pre-eclampsia (PE).

### 3.2. Patient Clinical Profiles at and after Transplantations

Of the lung recipients, 2.6% (3 patients) had a combined transplantation (one kidney and lung, the second, pancreatic islet cells and lung and the third, liver and lung). By comparison, a higher number of the kidney recipients underwent a combined transplantation (15.6%, *p* < 0.001), which, in most cases, was either pancreas and kidney, or liver and kidney, which was statistically significant.

The usual immunosuppressive treatment after either LT or KT surgery was a combination of mycophenolate mofetil (MMF), tacrolimus and corticosteroids (CTSs) (Table 2). After the transplantation (and before pregnancy if this occurred), 16.7% of kidney recipients experienced a rejection episode (that was mostly antibody-mediated), compared to 65.8% of lung recipients (*p* < 0.001) who experienced cell-mediated rejection (Table 2). The values at one and two years after SOT remained constant for both FEVI in the LT patients and GFR in the KT patients.

One lung and one kidney recipient developed gynecological cancers in the years following the procedure: a vulval epidermoid carcinoma and a breast invasive lobular carcinoma, respectively. Six patients (three in each group) developed cervical precancerous lesions that required conization. After the transplantation, close to 60% of all patients, whether LT or KT recipients, used progestin as their contraceptive method. Approximately 18% used a combined oral contraceptive (COC) with estrogen + progestin, and 16% and 22% of LT and KT patients, respectively, used barrier contraception.

### 3.3. Pregnancies

We identified 33 pregnancies in 24 women (11.4% of all 210 patients); 9/114 (7.9%) in lung recipients and 15/96 (15.6%) in kidney recipients. Of the 24 women, 17 were pregnant once, 5 were pregnant twice and 2 were pregnant three times (Figure 2). There was no statistical difference in the incidence of gravidity between the two transplantation groups (7.9% versus 15.6% for LT versus KT, respectively; *p* = 0.07).

Comparisons between patients who became pregnant post-SOT (*n* = 24) versus those who did not (*n* = 186) revealed no statistically significant differences for any of the maternal characteristics assessed, except for the incidence of diabetes, which was higher in the patients with no pregnancy (*p* = 0.04; Appendix A). Of note, all pregnant LT patients had cystic fibrosis (Appendix A). Of the KT pregnant patients, 33.3% had glomerulopathy, 20% had undetermined nephropathy, while none of the women with diabetes mellitus became pregnant.

No statistically significant differences were observed for graft functionality between the two transplantation groups as reflected by either glomerular filtration rate (GFR) for kidney recipients or timed forced expiratory volume (FEV1) for lung recipients (Appendix A). However, the values for both GFR and FEV1 were slightly improved 1 and 2 years after transplantations in the pregnant patient group (Appendix A).

The interval between transplantation and conception was similar for the two groups (3.2 ± 0.8 versus 4.2 ± 2.9 years for the LT and KT groups, respectively; *p* = 0.43; Table 3). However, the mode of conception was different: 100% were spontaneous for KT recipients versus 45.5% for LT recipients (*p* < 0.001), the remainder being medically assisted with ART plus ovulation induction. Most pregnancies were carefully planned for both SOT groups (77.3% and 72.7% after KT and LT, respectively) and underwent a switch of immunosuppressive medication mostly from MMF to azathioprine. Only four pregnancies began under MMF, none of which resulted in delivery: one ectopic pregnancy, one abortion, one medical termination and one ended in early miscarriage. In one pregnancy, MMF was switched to azathioprine shortly after the diagnosis.

No statistically significant differences were observed between the two groups for any pregnancy outcomes (Table 3), and no patient had a diagnosed graft rejection during pregnancy. Of the 33 pregnancies, 7 resulted in early miscarriage (21.2%), 1 resulted in a spontaneous abortion at 15 weeks gestation (due to chorioamnionitis after the pregnancy began under MMF treatment, which had been quickly switched to azathioprine), 1 was a medical termination (of an unplanned pregnancy that occurred after a chronic humoral rejection and a planned second lung transplantation), 1 was a mole, 1 was an ectopic pregnancy and 1 (which was undesired) underwent induced abortion. The livebirth rate was 63.7%: 21 pregnancies resulted in the delivery of a liveborn, with an equal incidence of delivery between the two groups (63.6% versus 63.7%, for LT versus KT, respectively).

Among those pregnancies >12 WG, 42.9% of LT and 53.3% of KT patients developed hypertensive disorders of pregnancy. One patient had gestational hypertension in the KT group, three had PE in LT group and seven had PE in KT group, which contributed to the increased rate of prematurity (Table 3). Indeed, of these patients, at least six had severe PE and two had a HELLP syndrome (one in each group).

Two patients in each group developed a severe infection during pregnancy. Three patients had chorioamnionitis at 15, 26 and 33 WG (the 2nd newborn died 15 days after birth) and one patient in the LT group had pyelonephritis, which required an intravenous antibiotic. Furthermore, one of the lung recipients had a significative decrease in FEV1 during pregnancy, which led to premature delivery at 34 WG. Likewise, two kidney recipients had kidney failure during pregnancy, both of whom had induced labor, one at 37 WG because of acute renal failure and the other at 35 WG (due to spontaneous decreased GFR at 38 mL/min/1.73 m^2^). There were two cases of threatened preterm births in the LT group (29th week for both). There was one case of threatened preterm birth, three cases of gestational diabetes, one preterm rupture of the membrane (36th week) and one anomaly of fetal heart rate (35th week) in the KT group.

The average weeks for delivery were 34.6 ± 2.3 and 33.6 ± 4 WG, for the LT and KT groups, respectively. One fetal malformation occurred in the LT group (a ventricular septal defect). The average birth weight of all neonates was 2049 g and the caesarian-section rate among all patients was 52.4% with no significant differences observed for either between the two SOT groups.

## 4. Discussion

This dicentric study was performed to provide a descriptive analysis of a cohort of patients under 40 years of age to determine the incidence and course of pregnancies after SOT. The principal findings were that slightly more than 10% of these women (11.4%) had at least one pregnancy after transplantation (9 of the 114 (7.9%) lung recipients and 15 (15.6%) of the 96 kidney recipients) with a high incidence of hypertensive disorders during pregnancy (50%). The mean term of delivery was 34 WG and C-section was performed in 52% of the cases.

The overall low rate of pregnancy in this transplant patient population may be related to their complex medical history. However, the medical characteristics and graft function were similar for women who conceived versus those who did not. This strongly suggests that in this population, a fear of pregnancy is greater than the desire for pregnancy, whether arising from the woman’s or the transplant physician’s perspective.

The high frequency of a previous history of pre-eclampsia in kidney recipients [17,18,19] and the knowledge of a short life expectancy in lung recipients affected by cystic fibrosis might explain this fear. Consistent with many published studies [9,20,21], the importance of multidisciplinary counseling for SOT patients planning pregnancy cannot be over-emphasized and was performed on more than 70% of the pregnancies in our survey.

Lung transplantation is often a necessary step in the current care of patients with cystic fibrosis, although the development of new therapies, such as CFTR (cystic fibrosis transmembrane conductance regulator) modulators, seem consistent with pregnancy [22,23] and may obviate the need for transplantation. This approach may raise the otherwise low incidence of pregnancy in cystic fibrosis patients. In a study conducted by Boyd et al., only 5.4% (64 among 1143) of women with cystic fibrosis reported at least one pregnancy [24].

Following the transplantation and under immunosuppressive medications, the majority of our patients used progestin as the contraceptive method of choice. There is no evidence regarding the contraceptive method that should be used among SOT recipients [25,26,27], and previous surveys on effectiveness revealed various results [28,29]. Progestin may offer a safer method than progestin with estrogen, especially in this comorbid population (diabetes mellitus, unbalanced lipid profile) under immunosuppressant therapy. However, further studies are required to confirm this possibility.

Gynecological follow-up is essential in SOT patients as immunosuppressive medications after SOT are often associated with cervical dysplasia and carcinoma [30,31]. It is therefore important to encourage regular screening by pap smear for all female recipients who undergo long-term immunosuppressive therapy [32]. As recommended by the American Society of Transplantation, it should be mandatory to promote the HPV vaccination in this population [33].

Regarding the optimal timing for pregnancy following SOT, a consensus conference held by the American Society of Transplantation in 2003 established that pregnancy is usually safe after the first year, providing that there is stable graft function and no rejection episode in the year before conception [9]. There are no official guidelines following a lung transplantation, but most of the limited literature recommends a lapse of two years after the procedure to ensure systemic stability (immunosuppression state, infection risk, respiratory function) before conception [34]. In our population study, the interval between SOT and authorization for pregnancy was longer, with an average of 3.2 years and 4.2 years for LT and KT patients, respectively. The shortest intervals for our LT patients was 2.4 years and 1.5 years for the KT group.

In the setting of pregnant SOT patients, the health and safety of offspring are clearly of paramount importance. MMF is one of the major immunosuppressants used after SOT, but it is associated with an increased risk of spontaneous abortion and facial malformation as others mycophenolic acid products [11,35]. Therefore, SOT women must use an efficient contraceptive method and must continue to use it for six weeks after stopping MMF. Despite this, the risk of unplanned pregnancies and MMF-associated fetal teratogenicity is real in SOT patients. In our study, we observed a high proportion (24.2%) of unplanned pregnancies, a rate consistent with the published results [36], and we identified one fetus with a malformation in an LT patient (but who was not under MMF).

All pregnancies in our KT recipients were spontaneous, whereas more than half of the lung recipients were conceived using assisted reproductive technologies (ARTs). Minimal data are available on medically assisted pregnancies among lung disease patients. Edenborough et al. suggested that patients with cystic fibrosis may have reduced fertility due to increased cervical mucus plugging and ovulation disorders [37], which may lead to ART. The use of ART in our LT population (54.5%) seems greater than previously reported by Thakrar et al.: 21% on 14 pregnancies [36]. However, this may be due to a high usage of pre-implantation genetic diagnosis (PGD), which is an adjunct technology of ART used to screen out embryos with a mutation, which in this case is the CFTR gene, responsible for cystic fibrosis. Despite most kidney recipients having restored spontaneous fertility post-transplantation, some of them may still need ART [38,39]. However, no KT patient in our study population used ART.

We observed an early miscarriage rate of 18,2% for LT recipients, which is consistent with the available data [40]. However, the incidence of miscarriages in our KT population (22.7%) was slightly higher than reported by other investigators [10,41], although similar to that in the global population [42].

The hypertensive disorders during pregnancy incidence of 42.9% in our LT recipients was slightly lower than the 50% hypertension rate previously reported [16,40]. However, the data are only available for seven LT patients in our study. Slightly more than half of our KT recipients (53.3%) exhibited this disorder, which was higher than the PE rate previously described for this population (5% to 45% [9,10,12,14,43,44]). The high overall incidence of PE in our patients may have resulted from immunosuppressive therapy as tacrolimus and corticosteroids have known nephrotoxicity [45], but can also induce hypertension [4].

Severe infections are common triggers of severe prematurity in solid organ recipients [14,40] due to induced immunosuppression, and in our case led to a late loss as well as a neonatal death.

None of our patients had a diagnosed rejection episode during their pregnancies. This is consistent with the absence of SOT rejections during pregnancy reported in the previous studies [9,40]. The careful balance of a woman’s health status and authorization of gestation could explain this observation: the delay to pregnancy following transplantation was more than one year, there was no case of recent rejection before pregnancy and almost all those pregnancies carried to term were planned. Moreover, pregnancy induces a global immunotolerance status and we assumed that our patients were regularly monitored for tacrolemia, which can greatly vary and is important to stabilize [46,47].

Deliveries were mostly premature due to induction for maternal reasons (mainly hypertensive disorders, organ failure and infection), as previously described in SOT patients [10,15,16]. Only one patient in each group gave birth after 37 WG and 23.8% of births occurred before 32 WG. The average week of delivery was 34 ± 3.5 WG. Aside from the one fetal abnormality, none of the neonates presented any particular adverse medical conditions. However, pregnancies following SOT are high-risk pregnancies that require regular and multidisciplinary follow-ups (at least with the organ transplantation specialist and gynecologist) in a level 2 or 3 maternity hospital. A low birth weight is largely explained by preterm birth, but it would be beneficial to add the regular monitoring of the uterine artery and umbilical artery Doppler to each obstetric ultrasound to rule out intrauterine growth restrictions in cases of high vascular risks.

We observed a high C-section rate in our study: 42.9% among the lung recipients and 57.3% in the KT group compared to the mean C-section rate in the two institutions (22%). Mastrobattista et al. reported C-section rates of 30% and 49–59% in the same groups [19]. However, it may be prudent to question the frequent use of C-sections as the mode of delivery because increasing the proportion of C-sections in SOT recipients has not been shown to decrease the incidence of maternal morbidity [48,49].

Achieving pregnancy and delivering a child after SOT raises several ethical questions [21]. Having children is a strong life goal for many, and the societal “exclusion” caused by a disease requiring SOT is reinforced further by being childless. This is a fundamental point that should be taken into account by the medical team. Of note, uterus transplantation is a new transplantation procedure [50], the sole aim of which is to enable a woman to carry her child. Allowing a woman to be pregnant must be balanced against the associated risks. Moreover, maternal support of the child must be considered if the patient has a predicted, shortened lifespan. In 2017, the median survival time following lung transplantations for cystic fibrosis was 9.9 years [5]; it was 19.2 years following a live donor kidney transplantation and 11.7 years after a deceased donor kidney transplantation [6]. Thus, counseling must include all the above issues, and more surveys should be performed to explore women’s feelings and perceptions concerning these important topics.

The strength of our study was the analysis of a large cohort of women of a childbearing age who underwent SOT during a ten-year period. The study was designed not only to investigate maternal and fetal outcomes in this vulnerable patient population, but also to determine the incidence of pregnancy in this cohort. This study was dicentric with data extraction performed from two major transplantation centers in France. Foch Hospital is the first French center to perform lung transplantations [2], with 82 procedures performed in 2019, and is one of the French reference centers of cystic fibrosis. Bicêtre Hospital is also a major kidney transplantation center in France. The two centers performed, respectively, 121 (Bicêtre Hospital) and 83 (Foch Hospital) kidney transplantations in 2019 [51]. Moreover, our study included patients with two types of transplantation procedures. Nevertheless, our study was not without its limitations. It was retrospective with missing data for several variables, which resulted in low numbers for analysis. The marital status and desire to have a child of all women in the whole cohort were unknown, as well as the rate of breastfeeding. The descriptive design with no control group precluded analyses to explore associations between outcome results and the impact of the SOT itself.

## 5. Conclusions

Pregnancies following a kidney or lung transplantation are feasible with loss rates comparable to those in non-transplanted women, although the risks of pre-eclampsia and prematurity are high. Physicians caring for women of a childbearing age who are SOT recipients should take into account their desire to conceive. When in good health and in the absence of recent rejection, planning a pregnancy for this population is reasonable. The risks of maternal complications (infection risk; hypertensive disorders, including preeclampsia) and fetal outcomes (preterm birth) should be carefully evaluated and considered to provide complete information for thorough counseling before proceeding with pregnancy. Multidisciplinary care, which includes the transplant surgery team and perinatologists, should be provided during the pregnancy, delivery and post-natal period.

## Figures and Tables

**Figure 1 jcm-11-04792-f001:**
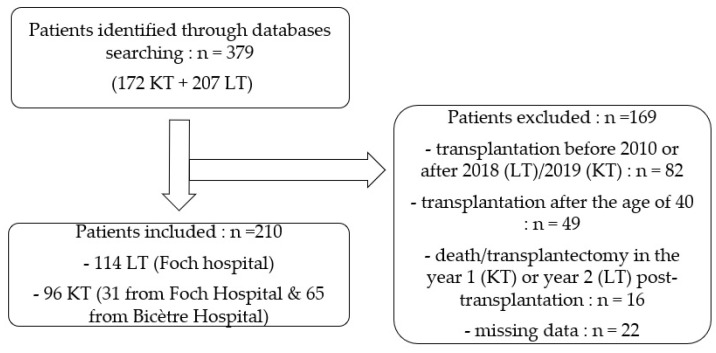
Flowchart.

**Figure 2 jcm-11-04792-f002:**
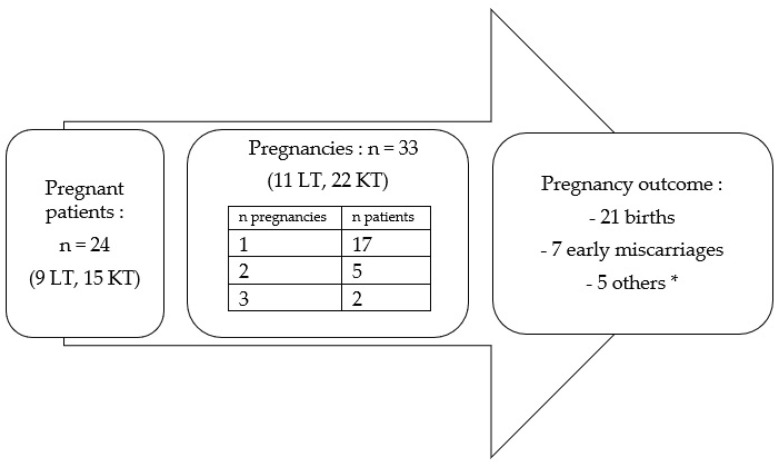
Characteristics of pregnancies in women after SOT. ***** 1 late loss, 1 medical termination, 1 ectopic pregnancy, 1 mole and 1 abortion.

**Table 1 jcm-11-04792-t001:** Demographic and medical histories of patients before transplantations.

	Lung Transplantation *n* = 114		Kidney Transplantation *n* = 96		*p*-Value
**Age at SOT, years**	**27.6 ± 5.59 ^1^**		31.1 ± 5.93		<0.001
BMI at SOT, kg/m^2^	18.0 ± 2.15		23.6 ± 5.39 (*n =* 79)		<0.001
History of					
Hypertension	3 (2.6%) ^2^		68 (70.8%)		<0.001
Diabetes mellitus	55 (48.2%)		19 (19.8%)		<0.001
Gyn. cancer	1 (0.9%)		0		1.000
Gravidity (G)					<0.001
G0	86 (75.4%)		47 (49.0%)		
≥G1	28 (24.6%)		49 (51.0%)		
Parity (P)					<0.001
P0	94 (82.5%)		53 (55.2%)		
≥P1	20 (17.5%)		43 (44.8%)		
History of pre-eclampsia	*n* = 9		*n* = 20		
(of n: ≥P1)	0		12 (60.0%)		0.003
Etiology	CF Bronch. Fibrosis Other ^3^ PCD	104 (91.2%) 4 (3.5%) 3 (2.6%) 2 (1.8%) 1 (0.9%)	Glom. Unknown Diabetes Interst. Vasc. PKD Other ^4^	25 (26.0%) 19 (19.8%) 13 (13.5%) 6 (6.3%) 5 (5.2%) 5 (5.2%) 23 (24%)	
Duration on dialysis before SOT, years	NA		*n =* 93 2.34 ± 2.23		

^1^ Values are mean ± SD. ^2^ Values are in parentheses are the percentages of the given number of patients in each transplantation group unless otherwise indicated. ^3^ Includes emphysema, bronchiolitis. ^4^ Includes urological malformation, side effects of use of calcineurin inhibitors, systemic lupus erythematosus, scleroderma, Alagille syndrome, acute tubular necrosis, tuberous sclerosis complex, hemolytic uremic syndroma, bilateral persistent obstruction, hyperoxaluria. BMI: body mass index, CF: cystic fibrosis, PCD: primary ciliary dyskinesia, bronch.: bronchiectasis, glom.: glomerulopathy, vasc.: microvascular condition for hypertension, PKD: polycystic kidney disease, interst.: interstitial kidney disease.

**Table 2 jcm-11-04792-t002:** Clinical profiles of patients at and after transplantations.

	Lung Transplantation *n* = 114	Kidney Transplantation *n* = 96	*p* Value
Multi-SOT	3 (2.6%) ^1^	15 (15.6%)	<0.001
IS therapy (year 1) by			
MMF	98 (86%)	84 (87.5%)	0.744
Tacrolimus	113 (99.1%)	92 (95.8%)	0.181
CTS	114 (100%)	93 (96.9%)	0.094
Other	17 (14.9%)	16 (16.7%)	0.728
≥1 T rejection	75 (65.8%)	16 (16.7%)	<0.001
1st rejection type:	*n* = 75	*n* = 16	
Antibody-mediated	6 (8.0%)	8 (50.0%)	<0.001
Cellular	64 (85.3%)	5 (31.3%)	<0.001
Combined	5 (6.7%)	3 (18.8%)	0.143
GFR ^2^ (mL/min/1.73 m^2^) 1 year after SOT 2 years after SOT	NA	59 ^3^ ± 23 (*n =* 83) 57 ± 24 (*n =* 72)	
FEV1 ^4^ (%) 1 year after SOT 2 years after SOT	81.1 ± 17.6 82.3 ± 18.2 (*n =* 112)	NA	
Gyn. cancer after SOT	1 (0.88%)	1 (1.04%)	1.000
Contraception after SOT	*n* = 81	*n* = 41	
Progestin	52 (64.2%)	24 (58.5%)	0.542
Copper IUD/condom	13 (16.1%)	9 (22.0%)	0.423
Estrogen + progestin	15 (18.5%)	7 (17.1%)	0.844
Other	1 (1.2%)	1 (2.4%)	0.621
Cervical conization after SOT	*n* = 25 3 (12.0%)	*n* = 12 3 (25.0%)	0.366

^1^ Values in parentheses are the percentages of the total number of patients in each SOT group unless otherwise indicated. ^2^ Normal GFR is ≥90 mL/min/1.73 m^2^ for a young woman, eGFR: estimated according to CKD EPI equation. ^3^ Values are mean ± SD. ^4^ Lower limit of normal FEV1 is 80% in the general population. Pregnancies were usually prohibited for women if the FEV1 was under 40%. T: transplant, IS: immunosuppressive, MMF: mycophenolate mofetil, CTSs: corticosteroids, FEV1: timed forced expiratory volume, gyn.: gynecologic, IUD: intrauterine device.

**Table 3 jcm-11-04792-t003:** Characteristics of pregnancies following transplantations.

	Total Pregnancies *n* = 33	Pregnancies in LT Group: *n* = 11 (114 Patients)	Pregnancies in KT Group: *n* = 22 (96 Patients)	*p* Value
Patients with ≥1 Pregnancy	24 ^1^ (11.4%)	9 (7.9%)	15 (15.6%)	0.079
Interval from SOT to authorization for 1st pregnancy (years)	*n* = 18	*n* = 8	*n* = 10	

	3.7 ± 2.16 ^2^	3.2 ± 0.76	4.2 ± 2.91	0.361
Mode of pregnancy				
Planned	25 (75.8%)	8 (72.7%)	17 (77.3%)	1.000
Unplanned	8 (24.2%)	3 (27.3%)	5 (22.7%)	
Mode of conception				<0.001
Natural pregnancy: *n* (%)	27 (81.8%)	5 (45.5%)	22 (100%)	
ART pregnancy: *n* (%)	6 (18.2%)	6 (54.5%)	0	
IS switch before				
pregnancy	27 (81.8%)	10 (90.9%)	17 (77.3%)	0.637
Outcomes				
Early miscarriages	7 (21.2%)	2 (18.2%)	5 (22.7%)	1.000
Medical termination	1 (3.0%)	1 (9.1%)	0	0.333
Other ^3^	4 (12.1%)	1 (9.1%)	3 (13.6%)	0.706
Births	21 (63.7%)	7 (63.6%)	14 (63.7%)	1.000
Complications	*n* = 22	*n* = 7	*n* = 15	
PE/HTA	11 (50.0%)	3 (42.9%)	8 (53.3%)	1.000
Infection	4 (18.1%)	2 (28.6%)	2 (13.3%)	0.565
Graft failure	3 (13.6%)	1 (14.3%)	2 (13.3%)	1.000
Rejection	0	0	0	
Fetal malformation	1 (4.5%)	1 (14.3%)	0	0.318
Other ^4^	7 (31.8%)	2 (28.6%)	5 (33.3%)	0.823
WG at delivery	*n* = 21	*n* = 7	*n* = 14	
All deliveries	34.0 ± 3.5	34.6 ± 2.3	33.6 ± 4	0.551
<32 WG	5 (23.8%)	1 (14.3%)	4 (28.6%)	0.624
32–37 WG	14 (66.7%)	5 (71.4%)	9 (64.3%)	1.000
Full term	2 (9.5%)	1 (14.3%)	1 (7.1%)	1.000
Weight at birth (g)	2049 ± 681 (*n* = 18)	1976 ± 121.2 (*n* = 5)	2076.9 ± 806 (*n* = 13)	0.788
C-section delivery	11 (52.4%; *n* = 21)	3 (42.9%; *n* = 7)	8 (57.1%; *n* = 14)	0.659

^1^ Values in parentheses are the percentages of the total number of pregnancies in each group, except where otherwise indicated. ^2^ Values are mean ± SD. ^3^ Includes late loss, ectopic pregnancy, mole and abortion. ^4^ Includes gestational diabetes, preterm rupture of membranes, abnormal fetal heart rate, threat of preterm labor. ART: assisted reproductive technology, IS: immunosuppressant, PE: preeclampsia, HTA: gestational hypertension, WG: week of gestation.

## Data Availability

Not applicable.

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
