# Peer review of "Pregnancies and Gynecological Follow-Up after Solid Organ Transplantation: Experience of a Decade"

_jcm, 2022, doi:10.3390/jcm11164792_

Round 1
Reviewer 1 Report
This is a manuscript that provides information on a major issue for women transplant recipients.
Although this is a retrospective study, the methodology remains valid and rigorous. The retrospective analysis therefore remains valid.
The inclusion criteria are solidly defined, allowing the study population to be well targeted.
The results are well presented and consistent.
The discussion is lucid and highlights the essential points.
Author Response
Thank you for your encouraging comments ! I think there is no point to answer for us.
Reviewer 2 Report
Thank you for the opportunity to review a valuable manuscript.
Introduction. The title of the manuscript directs to solid organ transplantation. Solid organ transplantation includes the transplantation of the kidney, liver, pancreas, heart, and lung. I suggest to add the information why the only 2 types of solid organ transplantation were chosen.
Line 61 - growth restriction - I suggest to add fetal...
Line 66 - adverse gynecological conditions. Please explain this definition (Gynaecological disorders? Gynaecological cancers?).
Materials and Methods. Please speify or change the definitions - preganacy authorization (line 100, confirmation or other), scheduled/unscheduled pregnancy (line 102-103, unplanned or other), mole (line -108, molar pregnancy or other0, avbortion (line 108, the term abortion includes and miscarriage, may be better medical abortion), pregnancies ere usually prohibited (line 124 - contraindicated).
Results. please check the lines 169-170. Twelve of the 20 patients (27.9%) 169 with known history and who had previously given birth had experienced preeclampsia. The numbers in the table 1 are different.
Contraception. Also I suggest to explain - what contraception do you mean "Progestin" - IUD with levonorgestrel, oral tablets of progestin, depo injection. Please explain why the copper IUD and condoms are included in the same group.
The 4 categories of hypertensive disorders of pregnancy are chronic hypertension, gestational hypertension, preeclampsia-eclampsia, and chronic hypertension with superimposed preeclampsia. I suggest to follow this in all manuscript.
Conception. Please specify medically assisted pregnancies (line 223) - ART or ART plus ovulation induction or other.
Lines 258 - 259. There were two cases of threatened preterm birth in LT group (29th week for both) and one threatened preterm birth. Please specify why it is so important - due to fetal lung maturation or other reasons.
Also I suggest to add the information in the parts Results or Discussion about the caesarean delivery rate in these two hospitals in general and in preterm delivery groups and compare with your study group results.
May be you have data on breastfeeding in your study group. It is important to discuss this in the manuscript too.
Discussion.
Line 357-358. Please specify the reasons.
I suggest to use better the definition - perinatologist instead of gynaecologist (line 59, 408).
It was a really great pleasure to review this manuscript, it contains a very useful information for clinicians.
Author Response
We would like to thank the reviewer for kindly accepting to review our manuscript. We feel that revisiting the manuscript according to the reviewer’s recommendations has greatly improved its readability.
Point 1: Introduction. The title of the manuscript directs to solid organ transplantation. Solid organ transplantation includes the transplantation of the kidney, liver, pancreas, heart, and lung. I suggest to add the information why the only 2 types of solid organ transplantation were chosen.
Response 1: Thank you for your comment. We explained in introduction why only two types of SOT were studied. It was due that only these two kinds of organs were transplanted in these two centers. Line 65-68: “The aim of the present study was to describe the mode of contraception used, the incidence of pregnancies, maternal and fetal outcomes, and adverse gynecological conditions in recipients who underwent SOT transplantation in two centers in France spanning a decade. In these centers, only lung and kidney transplantations were performed.”
Point 2: Line 61 - growth restriction - I suggest to add fetal...
Response 2: we added fetal as asked line 61
Point 3: Line 66 - adverse gynecological conditions. Please explain this definition (Gynaecological disorders? Gynaecological cancers?).
Response 3: Thank you for your suggestion. we clarified it, removed the term “adverse gynecological conditions” and replace it by “gynecological cancers” line 66.
Point 4 : Materials and Methods. Please specify or change the definitions - pregnancy authorization (line 100, confirmation or other), scheduled/unscheduled pregnancy (line 102-103, unplanned or other), mole (line -108, molar pregnancy or other0, abortion (line 108, the term abortion includes and miscarriage, may be better medical abortion), pregnancies were usually prohibited (line 124 - contraindicated).
Response 4: As asked, we replaced: scheduled/ unscheduled by planned/unplanned line 105, mole by molar pregnancy line 111, abortion by medical abortion line 111, prohibited by contraindicated line 128. We let the term pregnancy authorization line 100, because it is not confirmation of pregnancy but really the delay between the transplantation and the authorization to be pregnant (no medical contraindication like rejection or infection for example). To clarify this point we removed graft line 102 and replaced it by transplantation: line 102” interval between transplantation and pregnancy authorization”.
Point 5: Results. please check the lines 169-170. Twelve of the 20 patients (27.9%) 169 with known history and who had previously given birth had experienced preeclampsia. The numbers in the table 1 are different.
Response 5: We apology for this mistake it was 60% and not 27.8%. We corrected it line 173-174: “Twelve of the 20 patients (60%) with known history and who had previously given birth had experienced preeclampsia (PE).”
Point 6: Contraception. Also I suggest to explain - what contraception do you mean "Progestin" - IUD with levonorgestrel, oral tablets of progestin, depo injection. Please explain why the copper IUD and condoms are included in the same group.
Response 6: In progestin we included all contraceptions with progestin (IUD, oral and injections). We included copper IUD and condoms in the same group as a non-hormonal contraception. We clarified this point in methods line 99-101” Other data collected were use of contraception (classified into estrogen+ progestin; progestin: IUD with levonorgestrel, oral tablets of progestin and depo injection; non hormonal: Copper IUD and preservative and others)
Point 7: The 4 categories of hypertensive disorders of pregnancy are chronic hypertension, gestational hypertension, preeclampsia-eclampsia, and chronic hypertension with superimposed preeclampsia. I suggest to follow this in all manuscript.
Response 7: We replaced in the text: preeclampsia and gestational hypertension by hypertensive disorders of pregnancy (line 60, 105, 251, 276, 343, 411). In methods we explained the 4 categories of hypertensive disorders: line 105-107: “hypertensive disorders of pregnancy (including gestational hypertension, preeclampsia (PE) eclampsia, chronic hypertension and chronic hypertension with superimposed preeclampsia)”.
Point 8: Conception. Please specify medically assisted pregnancies (line 223) - ART or ART plus ovulation induction or other.
Response 8: It was with ART and ovulation induction. We specified it line 227-228: “the remainder being medically assisted with ART plus ovulation induction”
Point 9: Lines 258 - 259. There were two cases of threatened preterm birth in LT group (29th week for both) and one threatened preterm birth in KT. Please specify why it is so important - due to fetal lung maturation or other reasons.
Response 9: Thank you for this interesting remark. We don’t have really an explanation for this point.
Point 10: Also I suggest to add the information in the parts Results or Discussion about the caesarean delivery rate in these two hospitals in general and in preterm delivery groups and compare with your study group results.
Response 10: As suggested, we added in the discussion the comparison with the global rate of C-section in the two institutions. Line 372-373: “We observed a high c-section rate in our study: 42.9% among the lung recipients and 57.3% in the KT group compared to the mean c-section rate in the two institutions (22%).”
Point 11: Maybe you have data on breastfeeding in your study group. It is important to discuss this in the manuscript too.
Response 11: Unfortunately, we don’t have this information. We point this limitation in the discussion. Line 402-403: “The marital status and the desire to have a child of all women in the whole cohort are unknown, as well as the rate of breastfeeding.”
Point 12: Line 357-358. Please specify the reasons.
Response 12: We specified the reasons line 361-363: “Deliveries were mostly premature due to induction for maternal reasons ( mainly hypertensive disorders, organ failure and infection) , as previously described in SOT patients (10,15,16). »
Point 13: I suggest to use better the definition - perinatologist instead of gynaecologist (line 59, 408).
Response 13: We have made the correction
Reviewer 3 Report
Thank you for asking me to provide a review of this article, which has a subject of high interest nowadays, as all women, regardless they are in perfect health, or they underwent a complicated and of high risk operation such as Solid Organ Transplantation (SOT), deserve to concieve and have the opportunity to raise a child.
Regarding the structure and accuracy of the phrases, the manuscript has indeed well structured information, with supported evidence and well structured phrases. Being an, observational retrospective analysis, the study followed a group of 210 women, from which resulted a number of 33 pregnancies (21 livebirths and 7 misscarrigies), which may be quite sufficient for such a study. The aim of the analysis was to evaluate the type of contraception used, the incidence of pregnancies and maternal and fetal outcomes and adverse gynecological conditions in recipients who underwent a SOT. The strenghts of the study are the analysis of a large cohort of women of childbearing age who had a transplant during a 10-year period of time and also, the cohort included women with 2 types of transplantation procedures (lung transplant and kidney transplant).
The manuscript is original and well defined and so, the results provide an advance in current knowledge. The results are being interpreted appropiately and are significant, as well as are the conclusions, which are, of course, supported by the results. So the article is written in an appropiate way.
The study is correctly designed and the analysis is being performed at high standards, so the data are robust enough to draw the conclusion.
Surely the paper will attract a wide readership.
The English language is appropiate and well understandable and only has a few writting mistakes, which can easily be corrected, so that the article could be of highest quality.
I only have a few things to add in the lines below, strictly regarding the writting techniques:
· Line 18: have been, not „has been”
· Line 23: Progestiv, not „progestiv”
· Line 98: „.” after „circulation”
· Line 106: The investigated outcomes of the pregnancies were, not „Pregnancy outcomes investigated were”
· Line 155: transplantations, not „transplantation”
· Line 163: were presented, not „presented”
· Line 163: „,” after „profile”
· Line 163: compared to, not „compared with”
· Line 170: have experienced, not „had experienced”
· Line 175: statistically significant, not „statistically significantly”
· Line 294: „,” after „effectiveness”
· Line 308: „,” after „the procedure”
· Line 310: population study, not „study population”
· Line 321: we identified, not „identified”
· Line 323: „,” before „whereas”
· Line 337: „,” before „although”
· Line 346: triggers, not „trigger”
Author Response
Point 1: I only have a few things to add in the lines below, strictly regarding the writting techniques:
- Line 18: have been, not „has been”
- Line 23: Progestiv, not „progestiv”
- Line 98: „.” after „circulation”
- Line 106: The investigated outcomes of the pregnancies were, not „Pregnancy outcomes investigated were”
- Line 155: transplantations, not „transplantation”
- Line 163: were presented, not „presented”
- Line 163: „,” after „profile”
- Line 163: compared to, not „compared with”
- Line 170: have experienced, not „had experienced”
- Line 175: statistically significant, not „statistically significantly”
- Line 294: „,” after „effectiveness”
- Line 308: „,” after „the procedure”
- Line 310: population study, not „study population”
- Line 321: we identified, not „identified”
- Line 323: „,” before „whereas”
- Line 337: „,” before „although”
- Line 346: triggers, not „trigger”
Response 1: thank you verry much for your encouraging comments. We have made in the text all the corrections asked in red.